

# Comparing seasonal streamflow forecast systems for management of a fresh water reservoir in the Netherlands

Ruud T.W.L. Hurkmans[1,2], Bart van den Hurk[3], Maurice Schmeits[1], Fredrik Wetterhall[4], and Ilias G. Pechlivanidis[5]

[1]Royal Netherlands Meteorological Institute, de Bilt, the Netherlands
[2]HKV Consultants, Lelystad, the Netherlands
[3]Deltares, Delft, the Netherlands
[4]ECMWF, Reading, United Kingdom
[5]Swedish Meteorological and Hydrological Institute (SMHI), Nörrkopping, Sweden

**Correspondence:** Ruud Hurkmans (ruud.hurkmans@hkv.nl)

**Abstract.** For efficient management of the Dutch surface water reservoir Lake IJssel, (sub)seasonal forecasts of the water volumes going in and out of the reservoir are potentially of great interest. Here, streamflow forecasts were analyzed for the river Rhine at Lobith, which is partly routed through the river IJssel, the main influx into the reservoir. We analyzed multiple seasonal forecast data sets derived from EFAS, E-HYPE and HTESSEL, which differ in their underlying hydrological formulation, but are all forced with similar input from the ECMWF SEAS5 meteorological forecasts. We post-processed the streamflow forecasts using quantile matching (QM) and analyzed several forecast quality metrics. Forecast performance was assessed based on the available reforecast period, as well as on individual summer seasons. QM increased forecast skill for nearly all metrics evaluated. Particularly HTESSEL, a land surface scheme that is not optimized for hydrology, needed the largest correction. Averaged over the reforecast period, forecasts were skillful for the longest lead times in spring and early summer. For this period, E-HYPE showed the highest skill; Later in summer, however, skill deteriorated after 1-2 months. When investigating specific years with either low or high flow conditions, forecast skill increased with the extremity of the event. Although raw forecasts for both E-HYPE and EFAS were more skilful than HTESSEL, bias correction based on QM can significantly reduce the difference. In operational mode, the three forecast systems show comparable skill. In general, dry conditions can be forecasted with high success rates up to three months ahead, which is very promising for successful use of Rhine streamflow forecasts in downstream reservoir management.

## 1 Introduction

Lake IJssel in the Netherlands is a large fresh water reservoir. About 50% of the country's surface area is drained into the lake, and during summer droughts about 50% of the total fresh water use can be supplied from the lake (Waterman et al., 1998). The lake is fed by local precipitation and streamflow from a few rivers, of which about 85% is provided by the river IJssel, a distributary from the Rhine. The lake is drained by evaporation, water transport into the surrounding region for regional use, and streamflow to the Wadden sea. In 2018, the routine water level policy switched from a fixed seasonal cycle -with lake levels



of -0.40 m.a.s.l. in winter and -0.20 m.a.s.l. in summer - to a flexible lake level management (Rijkswaterstaat, 2018). This was, among other reasons, to anticipate possibly enhanced future water shortages (e.g., van den Hurk et al., 2014). By raising lake levels prior to, or early during a drought, an extra buffer is created. To optimally manage the lake level, early indications of

upcoming droughts are extremely valuable. Because the region recently experienced three dry years in a row (2018, 2019, 2020), attention for this subject has greatly increased.

In- and outflows of Lake IJssel are governed marginally by local meteorology, and predominantly by the IJssel's streamflow. The forecast skill of meteorological seasonal forecasts is known to be relatively low in Europe (e.g. Doblas-Reyes et al., 2013;

Yossef et al., 2017; Lucatero et al., 2018) due to the weak influence from large scale controls of variability such as ENSO, which is a dominant source of seasonal predictability in many other areas. Moreover, the North Atlantic oscillation (NAO) provides limited skill in Europe, and mainly in winter (Bierkens and van Beek, 2009; Sánchez-García et al., 2019; Scaife et al., 2014). On the other hand, forecasts of river streamflow generally have more skill, because it is not only determined by meteorology, but to a large extent by the system memory to initial conditions: groundwater, soil moisture, and snow (Crochemore

et al., 2020; Pechlivanidis et al., 2020). A correct estimation of the hydrological state at the beginning of the forecast, therefore, improves forecast skill (Arnal et al., 2018; Bierkens and van Beek, 2009). Because streamflow from the river IJssel is the main water source for Lake IJssel, a skillful seasonal streamflow forecast would substantially help water authorities to anticipate lake level adjustments. In 2018, for example, a decision to raise lake levels was taken when IJssel streamflow was already insufficient to do so (Rijkswaterstaat, personal communication), highlighting the need for seasonal predictions and decision-making.


A number of approaches exist to forecast streamflow at the seasonal scale. When, for example, streamflow is large determined by snow and relationships are well defined, regression approaches, based on data alone can be successful (e.g., Abudu et al., 2010). Other approaches employ (dynamic) hydrological models but are forced by observations, such as the Ensemble Streamflow Prediction method (ESP; Wood and Lettenmaier, 2006; Bennett et al., 2017), which assumes no skill in meteorolog-

ical forecasts and uses ensembles of historical meteorological forcing. Skill, therefore, only originates from initial conditions (Harrigan et al., 2018). However, meteorological forecast skill has improved over past decades, and currently a number of forecasting systems are available that takes also this source of forecast skill into account (e.g., Arnal et al., 2018; Pechlivanidis et al., 2020; Wanders et al., 2019). Most of these are based on meteorological seasonal forecasts from the ECMWF SEAS5 prediction system (Johnson et al., 2019) with different implementations of a hydrological model and the consequent process

representation.

The current study aims to analyze a number of forecast systems of the latter category. Its main objectives areto investigate: (1) whether streamflow forecasts provide meaningful information for reservoir management, (2) whether statistical postprocessing improves the forecast usability and (3) whether the available forecast systems provide similar information. In order to

answer these questions, we compare a number of seasonal streamflow forecasting systems. A statistical postprocessing method (quantile matching; QM) is applied to assess whether forecast skill can be improved. We compare the forecasts using a number





of forecast skill metrics by analyzing a long hindcast period, but also investigate specific years with low-flow conditions ( here defined as the years with the lowest average streamflow during the summer (April-September) half year. We express the streamflow forecasts in tercile classes and compare the corresponding statistics. Finally, to assess the differences between the employed systems and their significance we use the Diebold-Mariano significance tests (Diebold and Mariano, 1995).

The paper is organized as follows.In section 2, the study area is presented along with the available meteorological forecasts and hydrological forecasting systems. Section 3 presents the methodology for forecast postprocessing and evaluation. Section 4 presents the results, followed by a discussion in section 5. Finally, section 6 states the conclusions.

## 2 Study area, data and forecasting systems

### 2.1 Study area

Lake IJssel was cut off from open sea in 1932 and has been a fresh-water reservoir since then. Figure 1 shows the geography of the region including streamflow and intake locations for the surrounding regions. In dry periods, a large part of the Netherlands, approximately represented by light shading in Figure 1, depends on Lake IJssel for fresh water supply. One of the objectives to implement the flexible lake level management policy in 2018 is to allow raising water levels, and create extra buffers, when droughts are anticipated. The main source of water for the lake is the river IJssel, a distributary of the Rhine (Figure 1). Skillful forecasts of Rhine streamflow, therefore, are a valuable source of information to support lake level management.

During peak flows, about 11% of the Rhine streamflow is routed through the IJssel. Under drier conditions, this fraction can be increased by weirs in the Nederrijn. Figure 1 shows the relation between streamflow at Olst (in the river IJssel) and Lobith (in the Rhine upstream of the IJssel branch), averaged over June-October. On average in that period, about 15% of Rhine streamflow is routed through the IJssel, depending on specific conditions.

### 2.2 Data and forecasting systems

We analyzed streamflow (re)forecasts from three data forecasting systems driven by the ECMWF SEAS5 forecasts: streamflow derived from SEAS5, hereafter referred to as HTESSEL (Johnson et al., 2019), streamflow from the European Flood Awareness System, hereafter referred to as EFAS (Arnal et al., 2018) and streamflow from SMHI European seasonal forecasting service, hereafter referred to as E-HYPE (Pechlivanidis et al., 2020). From all systems, we extracted hindcasts and forecasts for the location Lobith, where the river Rhine enters the Netherlands (Figure 1). We consider hindcasts with 25 ensemble members for the period 1993-2015 and forecasts for the exceptionally dry summer of 2018, with 51 members. For all forecast systems, the daily time series are aggregated to weekly averages. A weekly time step is a relevant interval for low-flow events on one hand, but on the other hand leaves sufficient data points to allow a robust statistical analysis.





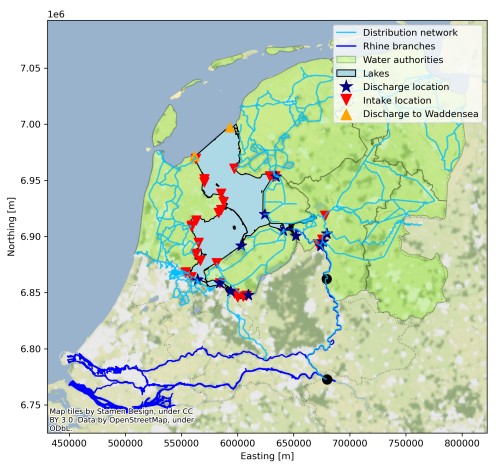
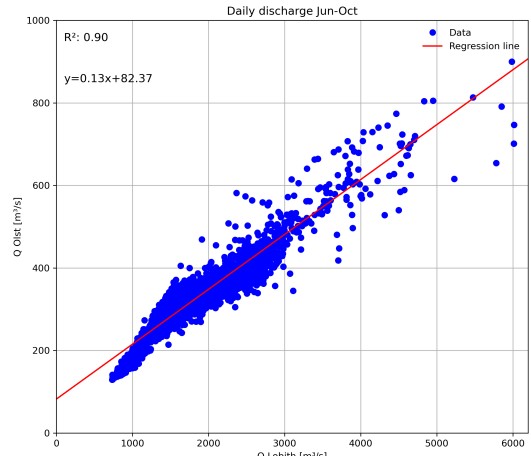

**Figure 1.** Left: geography of the region. Lakes IJssel and Marken are shaded in light-blue. Light-green areas indicate regions that (partially) depend on fresh water from Lake IJssel in dry periods with the main branches of the distribution network shown in light blue. Blue stars indicate inflow locations, red triangles extraction locations, and orange triangles discharge locations to the sea. Rhine branches are shown in dark blue. Right: the relationship between daily IJssel streamflow at Olst (the most downstream observation location in the IJssel) and daily Rhine streamflow at Lobith, for the period 1993-2018.

**Table 1.** Overview of the three forecasting systems.

| System | HTESSEL | EFAS | E-HYPE |
|---|---|---|---|
| Model | HTESSEL | Lisflood | HYPE |
| Resolution | 18x18 km$^{-2}$ | 5x5 km$^{-2}$ | 215 km$^{-2}$ |
| Variable | Subsurface+surface flow [mm] | Streamflow [$m^3 s^{-1}$] | Streamflow [$m^3 s^{-1}$] |
| Routing | No | Kinematic wave | Delay and dampening) |

### 2.2.1 HTESSEL

For the ECMWF SEAS5, we aggregated total runoff from the land surface scheme HTESSEL (Balsamo et al., 2009) at native resolution over the Rhine catchment. In this case, no surface routing is applied and also reservoirs are not considered. Because
90 we aggregate the daily time series to weeks, the lack of routing is assumed to have a negligible impact on the results. Different from the other systems, the land surface model HTESSEL has a focus on accurate representation of the water and energy balances, and is not tuned on simulating a correct river hydrograph. Especially during dry periods, the representation of river streamflow strongly relies on baseflow and this may be expected to be less accurate than the streamflow representation in the other models.



### 2.2.2 EFAS

In EFAS, the underlying hydrological model is Lisflood (Burek et al., 2013), a GIS based, distributed model running on a 5x5 km$^2$ resolution over entire Europe. It was calibrated on the period 1993-2002 at 693 European catchments (Arnal et al., 2018). Lisflood can represent snow, glaciers, frozen soils and soil moisture variability, which is parametrized using the Xinanjiang-formulation, similar to for example the Variable Infiltration Capacity (VIC)-model (Liang et al., 1994). Groundwater is included by means of a quick (shallow groundwater and macropore flow) reservoir and a slow (baseflow) reservoir, similar to the HBV model (Lindström et al., 1997). The resulting runoff is routed using a kinematic wave function through the river network, including reservoirs and lakes.

### 2.2.3 E-HYPE

The HYPE (HYdrological Predictions for the Environment) model is in its European application referred to as E-HYPE (Hundecha et al., 2016), is a semi-distributed model, where hydrological response units (HRUs) are based on land cover and soil. HRUs have an average size of about 215 km$^2$ . The soil is schematized using three layers, where the bottom layer accounts for groundwater. Deep aquifers are not taken into account, however both snow and glacier processes are. Streamflow is routed through the river network, where also reservoirs and lakes dampen the streamflow according to rating curves. The ECMWF SEAS5 meteorological forecasts are bias-corrected using the Distributed-based scaling method (DBS; Yang et al., 2010) and the HydroGFD dataset (Berg et al., 2018) as reference.

## 3 Methodology

### 3.1 Quantile matching

Quantile matching (QM) is a relatively simple approach to match the cumulative density functions (CDF) of forecasts and observations (Panofsky and Brier, 1968; Wetterhall et al., 2015; Crochemore et al., 2016; Ratri et al., 2019), since it has the advantage to correct the entire statistical distribution (Thrasher et al., 2012). In this application we compare all forecast values for a given forecast starting month and with lead times binned in 30-day intervals (where every lead time interval contains four weekly time steps). A specific month and lead time has, therefore, 92 (4 weeks per month times 23 years during the 1993-2015 period) data points. We then compose the observed and forecasted CDFs using 2% intervals and for each bin, we calculate a multiplicative factor.

We do not fit the multiplication factors to all years in the hindcast period, but employ a leave-one-out procedure; during the verification all metrics below show the average of all years, where the parameters for each year are fitted on all other years. This is justified because, for every calendar month, the autocorrelation of monthly averaged streamflow, with a lag of one year, is well below the significance level of 0.18 (based on an observed time series of 119 years, at a signifance level of 5%). For





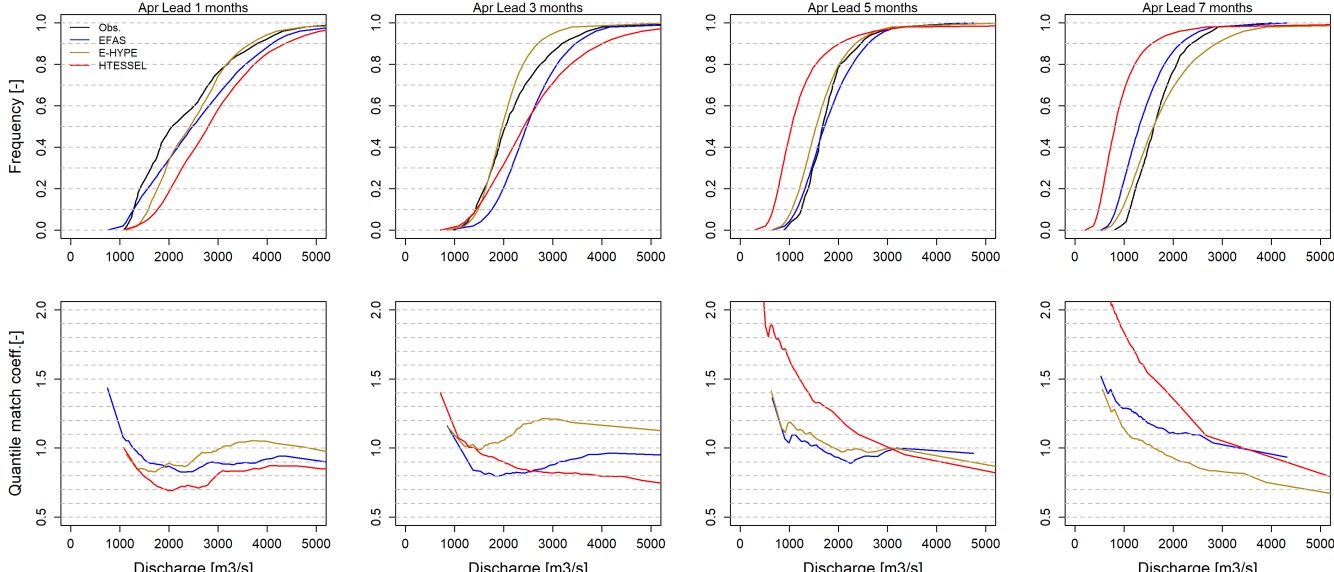

**Figure 2.** Forecast bias corrections using quantile mapping for forecasts starting on April $1^{st}$. Upper row shows CDF of EFAS (blue), E-HYPE (golden), HTESSEL (red) and observations (black) for lead times of 1, 3, 5 and 7 months where 1 indicates average streamflow over the first forecast month (in this case April). The lower row shows the resulting net multiplication factors of the bias correction.

illustration, Figure 2 shows the quantile matching for the forecasts starting on April $1^{st}$, and Figure 3 shows the multiplication factors as a function of lead time for the relevant forecast months.

From Figures 2 and 3 it is clear that HTESSEL, as expected, typically requires the largest correction. Also higher correction
coefficients are needed at lower streamflows, where all products typically underestimate low streamflows to some extent. In the next section we discuss the effect of applying QM on the forecast skill.

## 3.2 Forecast skill metrics

Forecast skill is evaluated in two ways. First, we analyze forecast systems averaged over the entire reforecast period (1993-
2015; section 3.2.1). In addition, the performance during specific years with varying streamflow conditions is investigated 3.2.2.

### 3.2.1 Reforecast period

We assess the forecast skill and how it is impacted by QM for the reforecast period (1993-2015). Three measures are used to illustrate this. Firstly, as a measure of general forecast performance, we use the Continuous Ranked Probability Score (CRPS),
which shows the deviation between observed and forecasted CDF. Secondly, to illustrate the effect of forecast postprocessing




**Figure 3.** Multiplaction factors as a function of lead time for six summer months (April-September) for EFAS, E-HYPE and HTESSEL.

using QM, the mean error (ME) of the ensemble forecasts medians is used; hence this is a deterministic metric. Thirdly, to focus on low-flow forecasts, we used the Brier score (BS) for underexceedance of lower tercile (33%) flows for each calendar month. All score metrics use observed streamflow as reference. We calculated the skill scores of CRPS and BS with the skill defined as:

$$S_{skill} = 1 - \frac{S}{S_{clim}} \qquad (1)$$


where $S_{skill}$ is the skill score, $S$ is the calculated score (CRPS or BS) and $S_{clim}$ is the score when using observed streamflow climatology as a benchmark forecast. If $S_{skill}$ is 1 the forecast is perfect, while negative skill implies lower performance than using climatology as a forecast system. In the following, the skill scores based on CRPS and BS are denoted as CRPSS and BSS, respectively. Similar to the QM-analysis, lead times are taken as 30-day intervals. In addition, we calculated the CRPSS





of precipitation from the ECMWF SEAS5 forecasts, averaged over the Rhine basin.

### 3.2.2   Individual years

To assess the forecast skill for individual events, we use tercile plots, which is a widely used approach to assess probabilistic forecasts (e.g. Doblas-Reyes et al., 2013; Johnson et al., 2019). Tercile plots show the probability that (in this case) monthly

discharge will fall in one of three categories: below-normal, normal or above-normal. The 33% and 67% percentiles (the terciles) form the boundaries between categories and were derived from the reforecast period. Associated to the tercile plots is the Ranked-Probability Skill Score (RPSS), which is essentially the Brier skill score generalized to more than two categories (Weigel et al., 2007), three in this case. Similar to CRPSS and BSS, RPSS is benchmarked using observed streamflow climatology as a reference.


In addition, a metric to indicate forecast resolution, which is the ability of the forecast to discriminate between subsequent events, is extracted from the terciles: the absolute difference between the probability of higher-than-normal versus lower-than-normal streamflow. For every calendar month, we bin the absolute value of this difference for all forecasts with 20% intervals. We refer to this value as the forecast resolution. Subsequently, we score for every bin how often the forecast was right, exclud-

ing cases where the observation category was 'normal'. The resulting metric is similar to a reliability plot, which shows the forecast probability versus the observed relative frequency.

## 4   Results

### 4.1   Impact of postprocessing on forecast skill

Here, we analyze the forecasts before and after postprocessing aiming to assess the impact of QM on streamflow forecast skill. Figures 4 and 5 show the skill scores (April-September) as function of lead tme for CRPSS and BSS, respectively. In addition, Figure 6 shows the mean error for the forecasts medians.

As can be seen in Figure 6, there is still significant bias present in especially raw HTESSEL and to a lesser degree in raw

EFAS. Note that meteorological input for E-HYPE has already been bias-corrected usiing the DBS method (Yang et al., 2010). However, as expected, QM effectively removes the bias in raw HTESSEL and EFAS and further reduces the bias in E-HYPE, with all systems performing similarly in all forecast months and lead times.

Similar conclusions are drawn for the probabilistic metrics CRPS and BS (Figures 4 and 5). Largest forecast skill is present

in spring and early summer, when streamflow appears predictable up to four months ahead. Presumably this is due to the

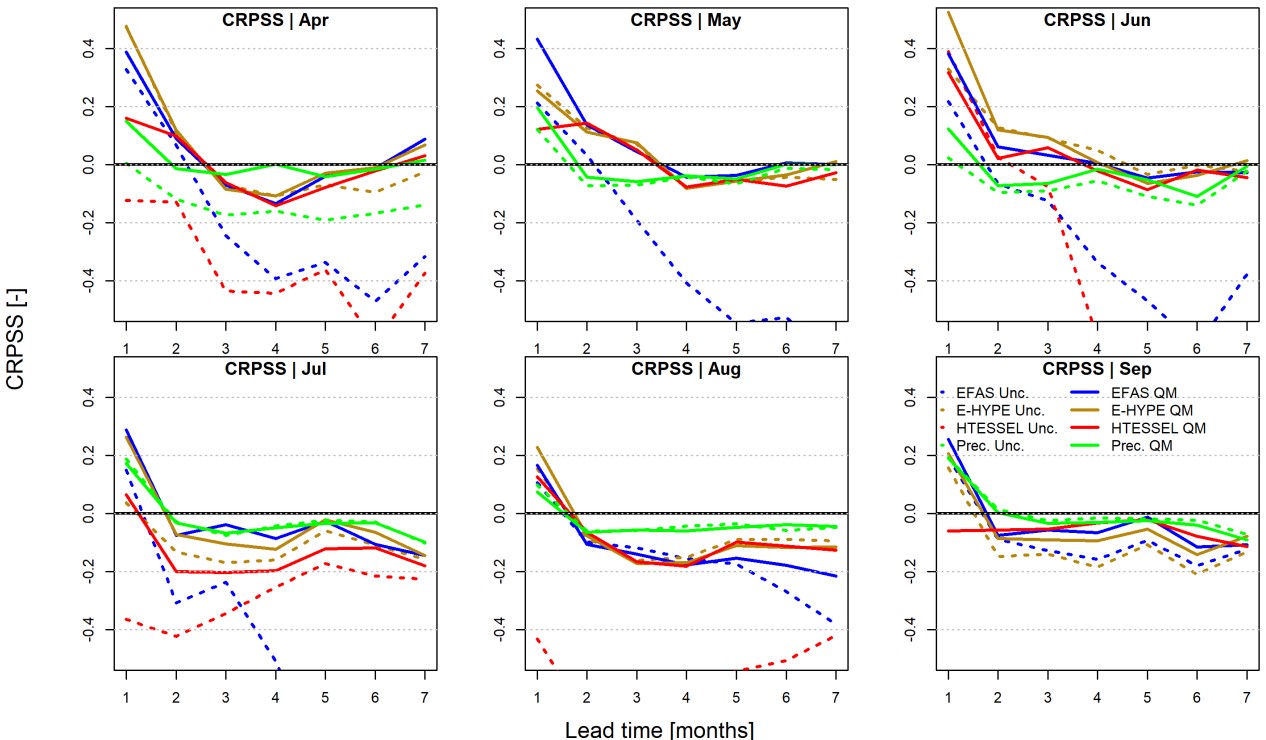

**Figure 4.** CRPSS for all hindcasts, with and without bias correction, for each lead time (months). Also shown is the forecast of precipitation aggregated over the Rhine basin (see text).

melting of the Alpine snow pack. Later in the summer, when streamflow is rain-dominated, the skill is much lower than in spring and gets negative after about 2 months. Regarding prediction system intercomparison, the differences between forecast systems are small: only in June E-HYPE shows slightly higher CPRSS-scores than EFAS and HTESSEL.

To analyze whether a specific forecast is better than another in certain parts of the year, we carried out a number of Diebold-Mariano tests (Diebold and Mariano, 1995) and assessed whether prediction errors of one forecast ensemble median are significantly lower (at $p < 0.05$) than another forecast median. Figure 7 shows a summary of the results per calendar month and lead time. We used one-sided tests, to be able to determine whether one forecast is significantly better or worse.

Figure 7 shows that bias-corrected EFAS and E-HYPE outperform bias-corrected HTESSEL throughout most of the seasonal
cycle. Before correction with QM, this difference is larger than post bias-correction (not shown). The differences between bias-corrected E-HYPE and EFAS are slightly more complex, with better performance for EFAS in spring (for short lead times) and E-HYPE in summer, which indicates the potential for multi-modelling in the region.





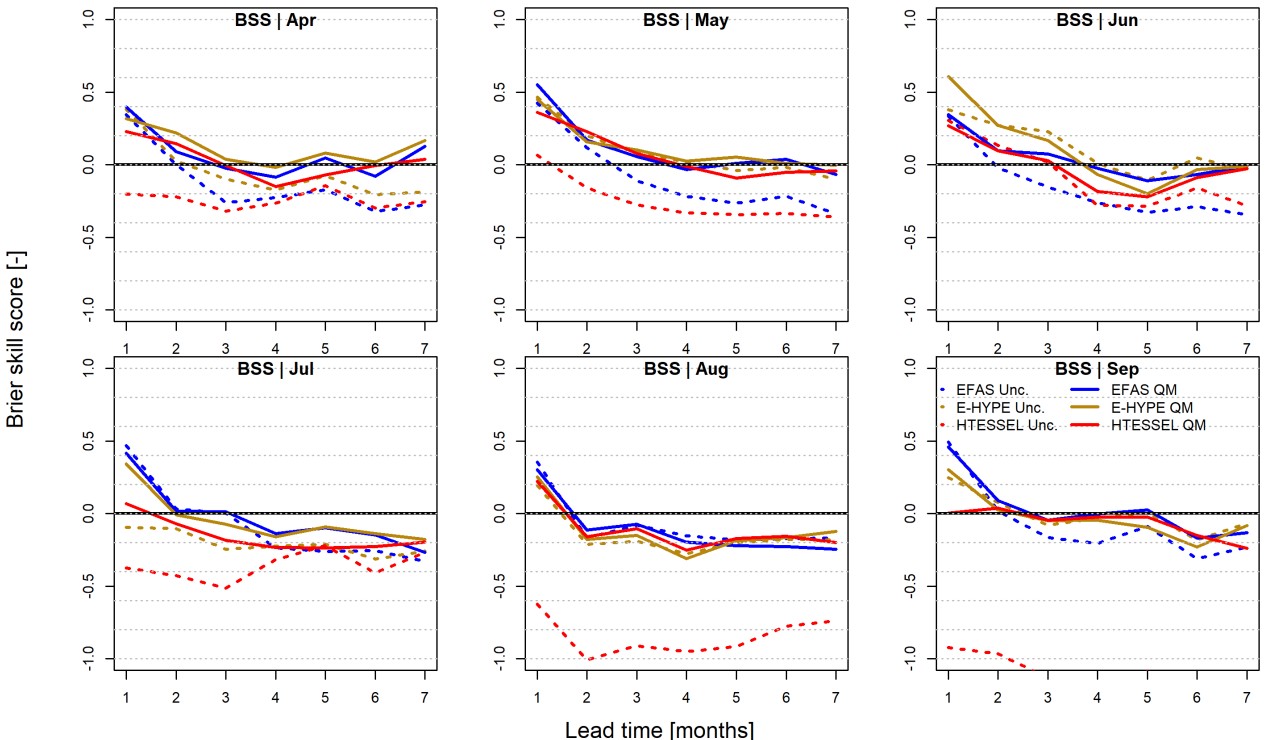

**Figure 5.** As Figure 4, showing the BSS. The threshold for calculating the Brier score is lower tercile ($33^{rd}$ percentile).

## 4.2 Assessment in specific years

Next, we assessed the streamflow forecast skill for selected "interesting" years from a decision-making perspective. As men-
tioned before, 2018 is highly relevant because a period with low Rhine (and IJssel) discharges coincided with a period of high
demand from the reservoir.

Figure 8 presents the forecasts at Lobith for a given month and lead time during 2018. We consider the bias-corrected results,
and carried out a leave-two-year-out validation and fit parameters based on the other years. Because at long lead times also
data from the previous year is used, we leave both the target and the previous year out of the fitting procedure. So, for example,
when considering the summer of 2003, QM-factors are derived on 2000, 2001, and 2004–2015.

Forecasts are promising for the summer of 2018; up to four months ahead lower-than-normal conditions were forecasted
with high probabilities, consistent with observations. To explore the forecast performance characteristics in other periods, we
ranked April through October averaged streamflow seasonal forecasts from the reforecast period (1993-2015) and 2018 and





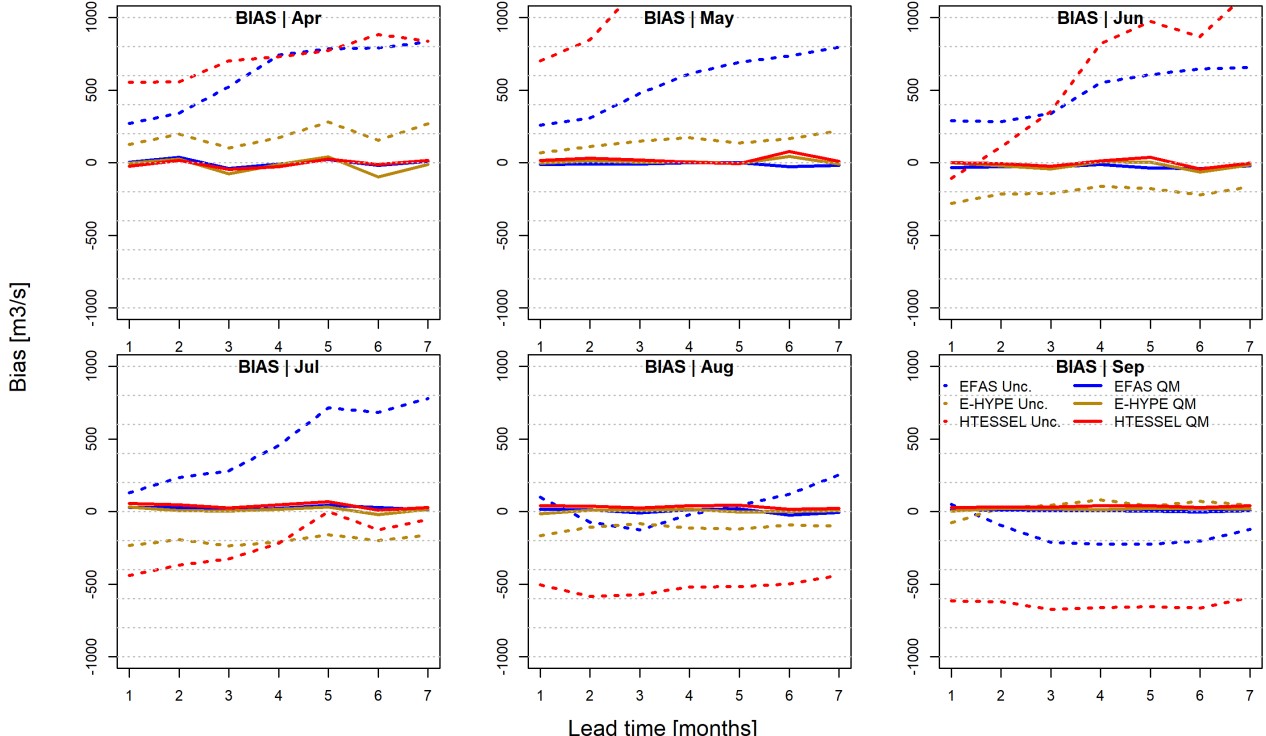

**Figure 6.** ME for all hindcasts, with and without bias correction, for each lead time (months).

selected the four years with highest and lowest average streamflow. Figure 9 shows time series of observed streamflow with ensemble medians of EFAS, E-HYPE and HTESSEL, both raw and bias-corrected for the forecasts initialized on April $1^{st}$, of the year with the highest (2001) and lowest (2003) streamflow from the ranking.

Figure 9 illustrates the impact of QM-correction. Especially during low streamflows, the HTESSEL forecasts severely underestimate observed streamflow highlighting the need for a high correction factor (Figure 2). During extremely dry summer seasons, such as 2003 and 2018 (the latter is not shown here), the raw HTESSEL forecasts are well in line with the observations. This explains the promising forecast performance as shown in Figure 8; however the raw HTESSEL forecasts also show low streamflow in normal and above-normal periods, which thus could lead to a high false-alarm rate. After bias-correction, the

differences between the forecast systems are relatively small. Figure 9 shows that after about three to four months, all forecasts converge to the climatological values, as expected. Figure 10 contains a graphical summary of the monthly RPSS-scores that are associated with the individual panels in the tercile plots (similar to Figure 8).

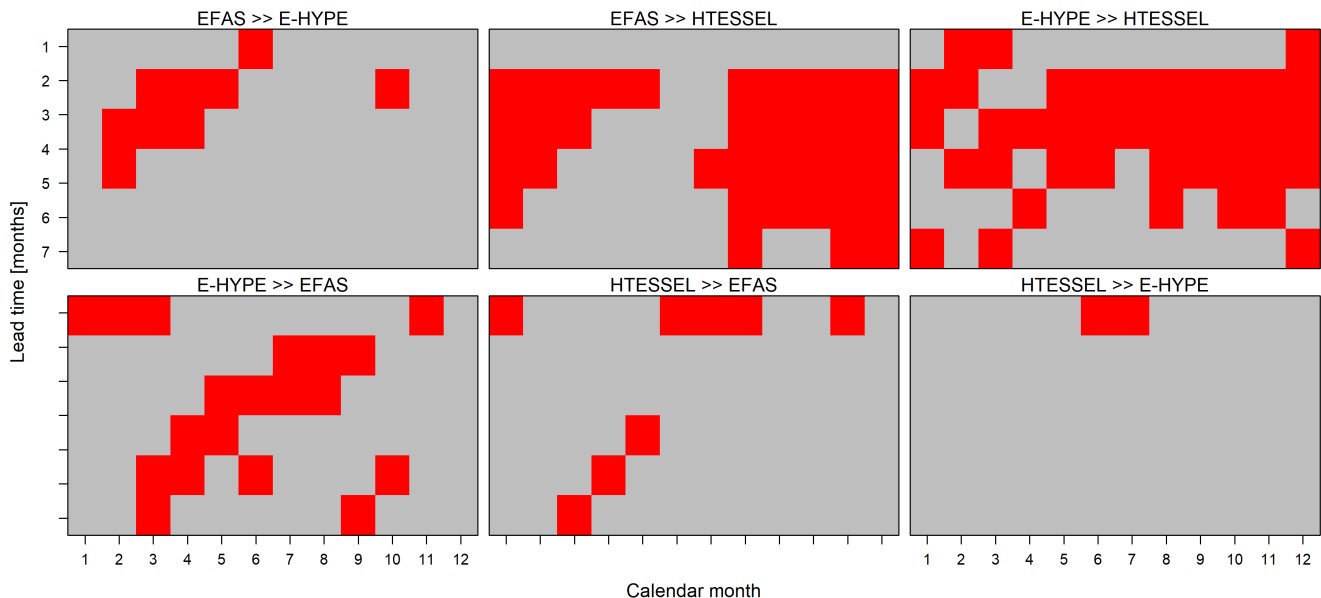

**Figure 7.** Summary of the results of Diebold-Mariano tests between bias-corrected forecast medians per calendar month and lead time. Significance of differences is plotted as a function of lead time (horizontal) and calendar month (vertical). The two rows show opposite tests: i.e., upper left shows where EFAS is significantly better than E-HYPE; lower left shows where E-HYPE is significantly better than EFAS. Significant differences are expressed by red colors.

In general, the RPSS values for three forecast systems are quite similar (Figure 10). Also it shows that the long-lead forecasts

for August and September 2018 have good performance. However, in years with extremely high or low streamflow, forecast skill seems to be above average (in relevance to Figure 4). To better illustrate this, Figure 11 shows the averaged RPSS over lead times up to four months and the summer half-year plotted against rank numbers of years ordered from years with lowest (2003, left) to highest (2001, right) streamflow, averaged over the period April-October.

In some cases, the tercile categorization amplifies relatively small differences in absolute streamflow, occasionally resulting in a considerable decrease of the monthly RPSS score. For example, in August 2011, the negative RPSS-scores are related to a temporary increase in observed streamflow at the end of July or beginning of August, which causes to shift the observed condition to 'Normal', whereas still lower than normal streamflow was forecasted.

The negative RPSS-values in Figure 10 are all associated with such events, in which observed streamflow is categorized in the middle tercile. From this visual inspection, there is some information in the difference between the probability of higher-than-normal versus lower-than normal streamflow (although this is not always reflected in the RPSS score). We present this





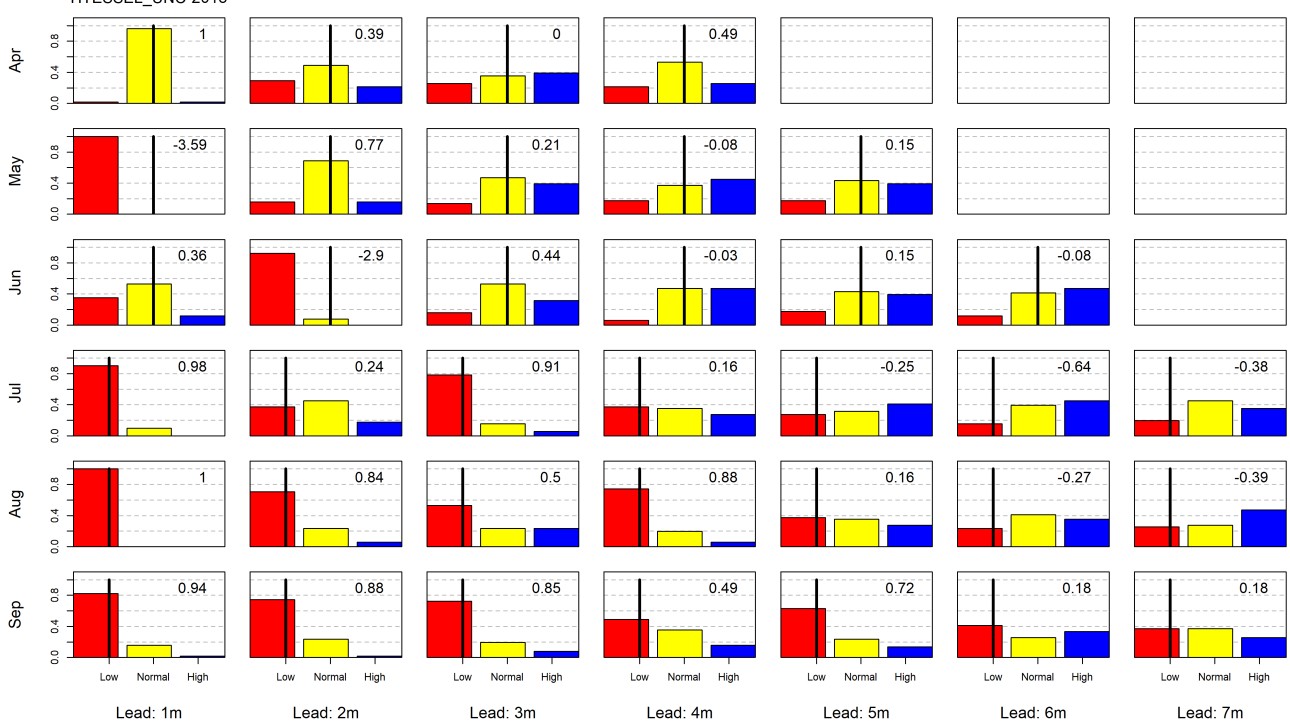

**Figure 8.** Example of a tercile plot using the raw HTESSEL forecasts at Lobith, for 2018. Red bars indicate below normal streamflow, blue bars above normal. The black line is the observed realization, the number indicates the RPSS, consistent with the thresholds and using observed climatology as a reference (Eq. 1). The bars show the probabilities for the three categories. Panels from top to bottom show the calendar months of 2018, with lead time (aggregated to 30-day bins) in horizontal rows.

information by plotting the fraction of correct forecasts as a function of forecast resolution, for summer months for each of the three forecast systems (Figure 12), according to the approach described in section 3.2.2. For comparison, also the reliability
plot (based on a binary forecast of under-exceedance of the lower tercile) is shown.

Due to relatively small datasets, the reliability plots are noisy; however they confirm the general conclusion that forecasts from all three systems are better in spring, and that E-HYPE generates more accurate forecasts in the summer months, (also shown in Figure 7). We note that the information as presented in Figure 12 may be useful for operational managers, where
instead of an average skill score, the probability for forecasts being correct is given. With the forecast archive growing, the noise in the results will decrease as the curves become more stable.

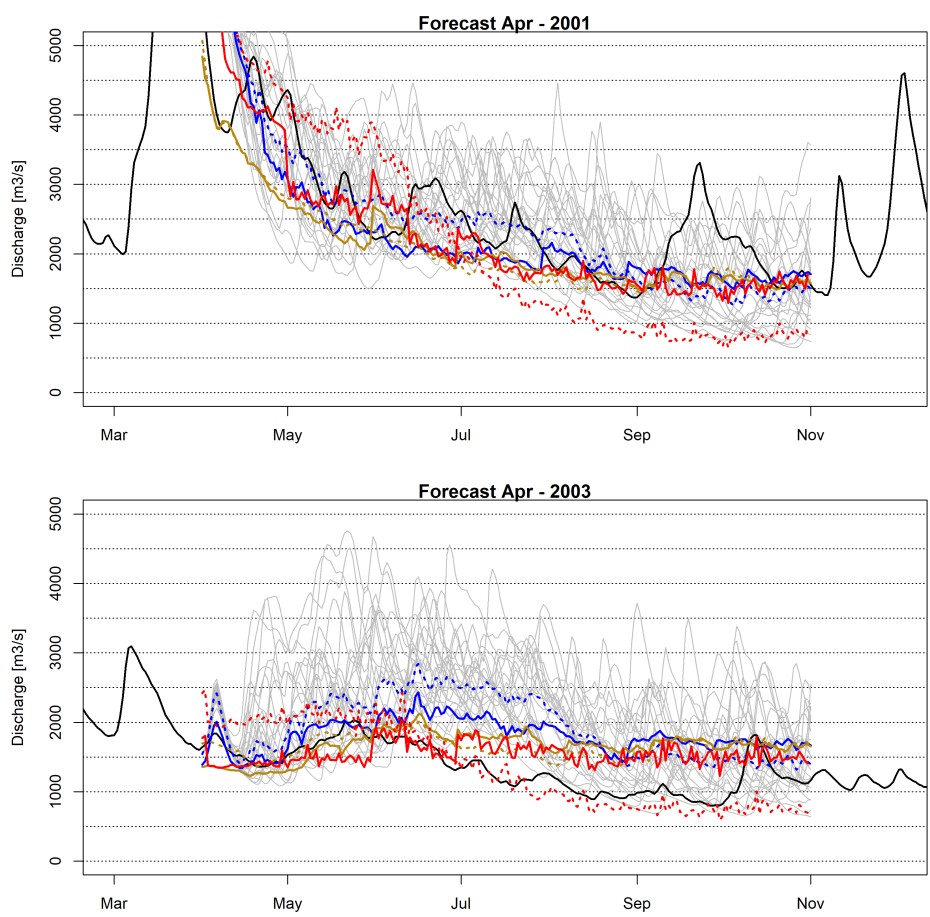

**Figure 9.** Example of forecasts from April $1^{st}$ for the year with the highest (2001; upper panel) and lowest (2003; lower panel) average summer streamflow. Observations are shown in black; ensemble medians of EFAS in blue; E-HYPE in golden, and HTESSEL in red. Solid lines refer to QM-corrected forecasts, while dotted lines refer to raw values. The thin grey lines indicate the ensemble members.



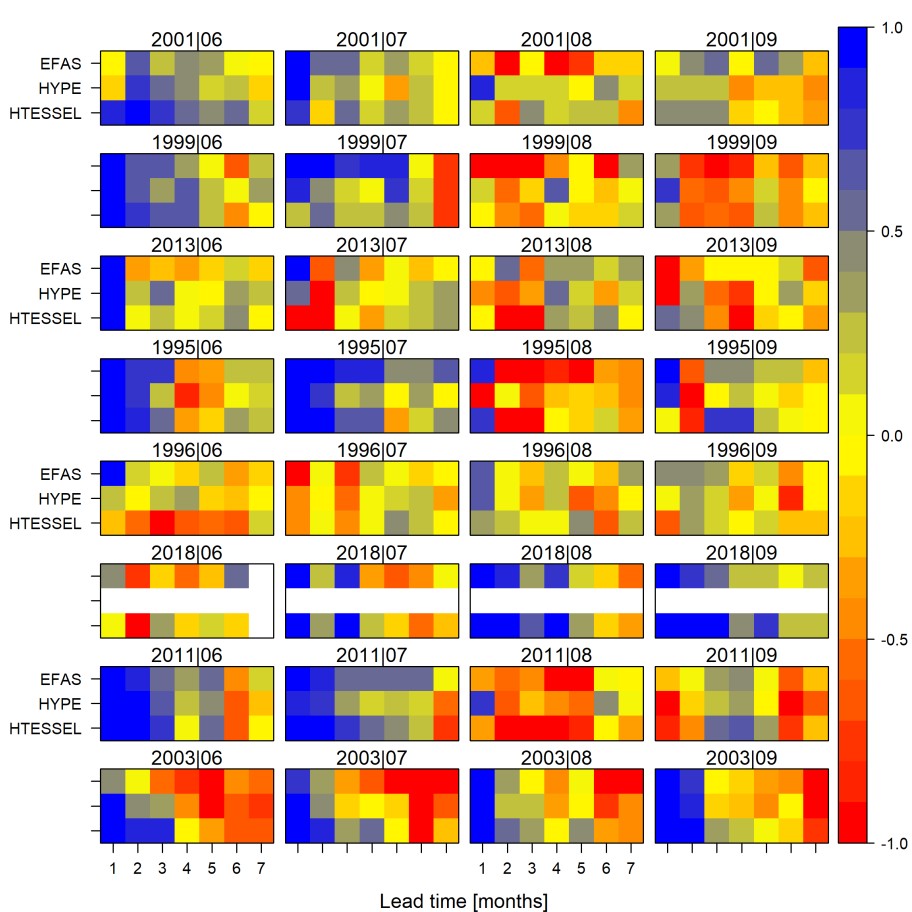

**Figure 10.** RPSS scores of QM-corrected monthly streamflow forecasts of EFAS, E-HYPE and HTESSEL. The panels show, from top to bottom, the four years with highest (four upper panels) and lowest (four lower panels) average summer streamflow. From left to right, the calendar months June-September are shown. Blue colours indicate skill.





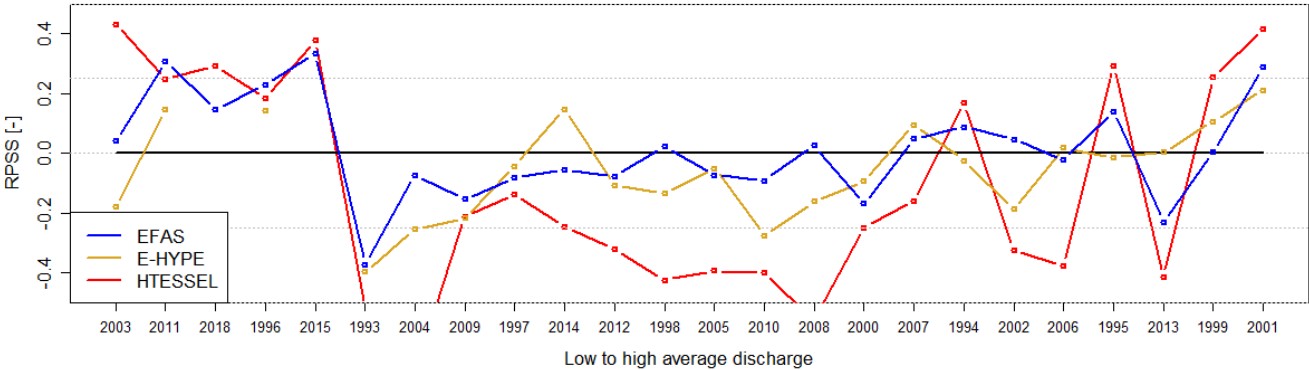

**Figure 11.** RPSS-scores, averaged over the summer half year (April-October) and lead times up to four months for years ranked from low (left) to high (right) averaged streamflow.

## 5 Discussion

### 5.1 Limitations in experimental setup

To assess the employed forecasts, we have selected a number of skill metrics, some of which depend on specific threshold values. For instance, the BSS was calculated with a threshold of 33% to be consistent with the terciles, which are commonly used in the dissemination of seasonal streamflow forecasts. We also calculated the BSS based on thresholds of 10% and 20%. Results were similar, although included noise as the number of underexceedences of the threshold decreased. Related to this, is the extent of the hindcast dataset. With the period 1993-2015 available, and metrics computed for each calendar month and lead

time, the number of data points is relatively small, as was also noted by Arnal et al. (2018), particularly when one assesses the extremes. When the reforecast dataset is extended, or combined with forecasts, the skill scores could become more stable. To optimize this number here, we have analyzed lead times in monthly intervals, with streamflow aggregated over weeks. Weekly aggregates are appropriate for typical low-flow events of the river Rhine (e.g. Hurkmans et al., 2010).

In this study, we focused on streamflow at Lobith as this location represents a (relatively) large catchment area. In the current application we use a fixed relation with IJssel streamflow based on historical data (Figure 1). In dry conditions, the fraction of Lobith streamflow routed over the river IJssel may be managed by weirs in the other branches (van Malde, 1988). To accommodate for this, machine learning algorithms may be able to include this management aspect in the relationship.

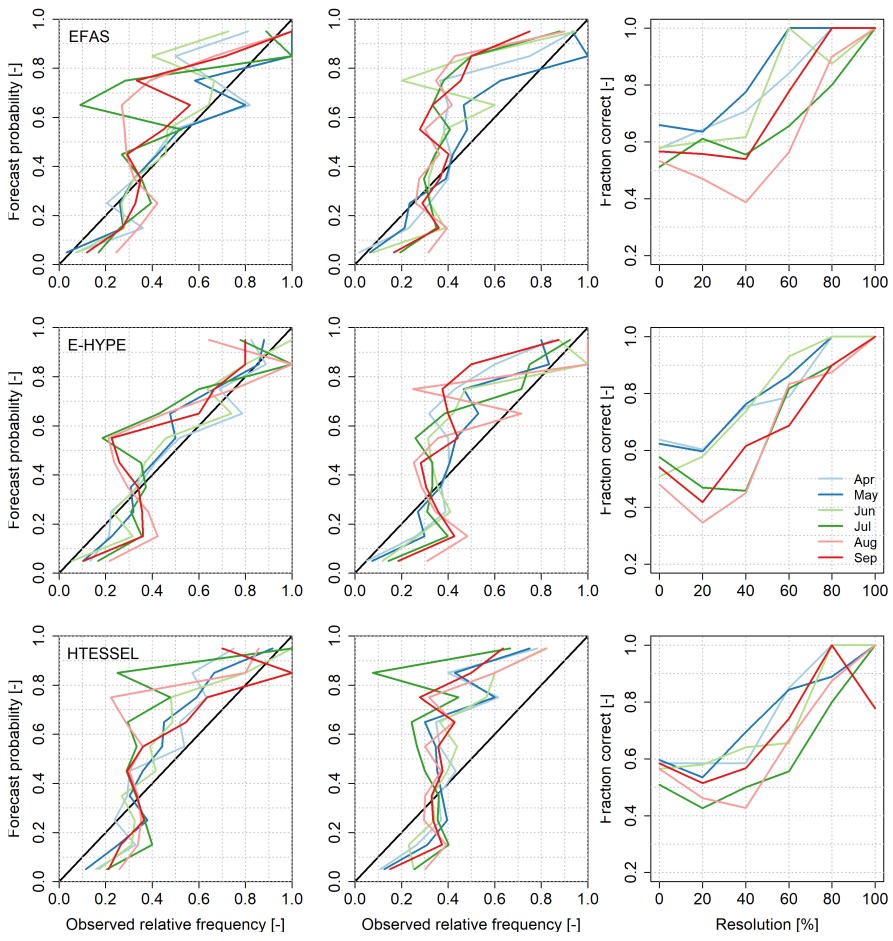

**Figure 12.** Forecast reliability plots for streamflow based on EFAS, E-HYPE and HTESSEL not exceeding the 33% lower tercile value (left columns) and exceeding the 67% upper tercile (middle column) for different calendar months. The right column shows the fraction of correct forecasts as a function of the forecast resolution.





## 5.2 Moving forward and service implementation

The other components of the reservoir water balance apart from streamflow, i.e., precipitation, evaporation and regional water extractions, depend on local meteorological conditions and are therefore less predictable. Ongoing research pertains to both improving the relationship between Rhine and IJssel streamflow and assessing how streamflow forecast skill propagates through the lake water balance in the lake level forecast skill.

The differences between the forecast systems, especially E-HYPE and EFAS, are difficult to disentangle without examining the stores and fluxes within the underlying hydrological models. It would be interesting to carry out such an exercise, for example with to objective to develop a multi-model forecast ensemble (Muhammad et al., 2018; Wanders et al., 2019). Recently, a high-resolution, distributed model specific for the Rhine basin has been developed (Imhoff et al., 2020). This has a number of features that might increase forecast skill, such as a more detailed representation of ground water and soil moisture and upstream surface water reservoirs. Producing seasonal forecasts based on this model and assessing whether forecast skill indeed improves are included in ongoing research.

Along with this future research a prototype of a (pre-)operational system is being developed. Recently, much effort has been put into bridging the gap between science and operational use (e.g., Soares et al., 2018; van Den Hurk et al., 2016; Lavers et al., 2020). Skillful hydrological forecasts are the first step in successful operational use. In addition, the ability of end-users to interpret the ensemble probabilities and make optimal decisions is needed (Giuliani et al., 2020). To achieve this, operational systems and their interfaces are best created in cooperation between developers and end-users (Golding et al., 2019). Development of the prototype will, therefore, be in close cooperation between the developers and Rijkswaterstaat.

## 6 Conclusions

We compared various metrics of forecast skill for a number of streamflow forecast systems for the Rhine catchment, with a focus on low-flows and monthly streamflow values. The investigated forecast systems were EFAS, E-HYPE and HTESSEL. All generate operational hydrological seasonal forecasts for entire Europe, hence their usability for regional applications can be explored. All three systems are forced by ECMWF SEAS5 seasonal forecasts and mainly differ in the underlying hydrological model, whilst in the case of E-HYPE the meteorological forecasts are first bias-corrected.

We show that streamflow forecast skill scores are high up to four months ahead in spring, when streamflow is dominated by snow melt, while the length of the skilful period decreases to about 1-2 months in summer when streamflow is mainly driven by rainfall. We also noted a relationship between the forecast resolution and skill. Moreover, both the forecast skill and the lead time with positive skill increase with the extremity of the hydrological event. From an operational perspective, when forecasts have a high resolution at long lead times, the probability of an upcoming anomalous event is relatively high, which is a relevant





notion for water managers.

After postprocessing the streamflow forecasts using a bias-correction, based on quantile matching, the difference between the three systems is small. In addition, the categorization of streamflow into terciles reduced the difference between the forecast systems. Consequently, we concluded that all bias-corrected forecast systems were able to provide useful information for reservoir operations. That was not the case for raw forecasts, since bias-correction proved to be essential. Depending on the season and lead time, there were differences between the forecast systems. To combine the strengths of all models and accu-
rately represent forecasted uncertainty, a multi-model ensemble is proposed for operations, in which depending on the season, one forecasting system could get more weight than the others. Finally, we note that the forecasts will be implemented in an operational reservoir management system and the usefulness (in terms of economic benefit and general decision-making) in managing reservoir water levels will be assessed.

*Code availability.*    R-scripts that were used for the analyses and the underlying datasets are available upon request.

*Author contributions.*    RH, BH and MS designed the study. IP and FW prepared datasets that were used. RH perfomed the analyses with input from BH and MS and prepared the manuscript with input from IP and FW. All authors contributed to revision of the manuscript.

*Competing interests.*    The authors declare that they have no conflict of interest.

*Acknowledgements.*    This research was financially supported by NWO project SWM-EVAP (ALWTW.2016.049). The EFAS seasonal fore-
casts are produced by the EFAS computational center in support of to the Copernicus Emergency Management Service(EMS) and Early Warning Systems (EWS) 198702.



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
