# Peer review of "Comparing seasonal streamflow forecast systems for management of a fresh water reservoir in the Netherlands"

_Hydrology and Earth System Sciences, 2021_

## Referee Comment (RC1)

**Review for HESS-2021-604: Comparing seasonal streamflow forecast systems for management of a fresh water reservoir in the Netherlands**

In general, the study adresses a very recend and highly relevant topic as it discusses the skill of seasonal forecasts for predicting runoff. Besides the "raw" runoff forecasts from HTESSEL, E-HYPE and EFAS, the authors also apply a simple but effective quantile mapping bias correction in order to match the distribitions between observed and forecasted runoff. The bias-corrected forecasts are then compared against observed runoff at a single gauge in the river Rhine at Lobith, that serves as a proxy for inflow to the river and lake IJssel. The authors conclude that after bias correction, streamflow forecasts have skill up to four months ahead in spring, when streamflow is dominated by snow melt. During summer, this skillful period decreases to about 1-2 months. While the study focuses on a single basin, the results are still important for the hydrological community.

**Major comments**

- But this limited scope is my main poin of criticism. As far as I can tell, the authors did not run any of the models, but rather used publicly available data. Furthermore, there are multiple data sources for runoff observations across Europe. Thus, while the study is certainly of high relevance for the water management of the IJssel reservoir, it could serve as a state-of-the-art skill assessment for a much larger domain and, hence, would be an update for, e.g., Arnal et al. (2018). None of the applied methods are tailored to the study domain or require any local adjustments or knowledge. Thus, the whole framework could be easily applied to multiple sites and, hence, present the skill of seasonal streamflow forecasts from current hydrological forecasting systems for major European river basins. This would be a very important and much needed contribution to the whole European hydrometeorologcal community. Therefore, I would suggest to include at lease some more rivers and gauges to a) increase the importance and scope of the study and b) see if the author's findings can be transferred to other river basins.
- Furthermore, other climatic variables like precipitation or temperature are equally important for the reservoir management. If the authors decide to stick to the IJssel reservoir, I would strongly suggest to include some other variables as well to make the whole study a bit more comprehensive.
- While this is a quite frequent comment, I also think that the overall language, wording and presentation of the paper could be substantially improved.

**Minor comments**

- Line 61: Add whitespace before "In section 2..."

- Lines 88 - 89: Maybe re-phrase to: Total runoff from the HTESSEL land surface scheme is aggregated at native resolution over the Rhine catchment?

- Line 89: Could you give a reference for this statement? Could the lack of a routing scheme explain the bad results of the raw HTESSEL forecasts?

- Line 99: similar to, for example, the Variable...

- Line 104: The HYPE-model, whose European application is referred to as E-HYPE, is a semi-distributed...

- Line 106: Whitespace after 215km^2.

- Line 125: This is the first time that you discuss the runoff observations that are used in the study. Please add some more details like the source of runoff observations to, e.g., section 2.2.

- Lines 149 - 150: How did you compute the CRPSS for precipitation forecasts? Which "reference" did you use?

- Lines 156 - 158: Remove "three in this case" as this is obvious from the previous sentences.

- Line 158: ...is benchmarked using AN observed streamflow climatology...

- Lines 161 - 163: This sounds extremely complicated and, to some extend, different to the general definition of the term "forecast resolution". But this might be unintentional and could be improved by simply citing standard references for the evaluation of forecasts.

- Line 171: lead tme

- Line 175: usiing

- Lines 185 - 192 and Figure 7: I do not know how "substantial" this Diebold-Mariano-Tests really are to your overall asssesment... Especially as you are comparing ensemble medians and, hence, do not consider the full ensemble. I assume that a simple comparsion of forecast-tailored skill-scores (e.g., a generalization of Figure 10) should give more comprehensive information.

- Line 201: ...QM-factors are derived DURING 2000, 2001, and 2004 - 2015

- Line 258: Why are you particularly referring to machine learning approaches here?

- Section 3.1: The method is ussually known as "Quantile mapping". Please correct this throughout the manuscript. Furthermore, how did you treat the tails of the distribution? Getting these parts of the distributions right is quite important especially as you are also focusing on low-flow events in your evaluation.

- Section 3.2: Maybe replace "Forecast skill metrics" with "Evaluation of forecast skill"?

- Section 3.2.1: Why did you only use the BS for low-flow forecasts? The skill of high-flow forecasts is quite important as well.

- Section 5: You do not really "discuss" your findings here, but rather summarize your study and give an outlook. As there are already several publications on this topic (e.g., Arnal et al., 2018; Samaniego et al., 2019; Ionita & Nagavciuc, 2020), it would be interesting to put your findings in the context of these other studies.

- Figure 1: What are the black dots? Increase fontsize of legend, R^2 and y --> hard to read

- Figure 2: Please increase font-size for the legend. Furthermore, I would remove "Forecast bias corrections using...." and rather just write "Upper row: CDFs from EFAS, E-HYPE, ... derived from

forecasts issued on April 1st for lead months 1 (April), 3 (June), 5 (August) and 7 (October); bottom row: mapping factors between observed and forecasted CDFs".

- Figure 4: Do you evaluate the skill of Apr. to Sept. forecasts from different issue dates (months) or rather the skill of forecasts that have been issued from Apr. to Sept.?

- Figure 8: You write that you're using raw HTESSEL-forecasts but in the text (Lines 198 - 201), you write that "we consider the bias-corrected results". Or did I understand something wrong here?

- Figure 9: Increase the "thickness" of the observations as they are very hard to distinguish from all other lines. Furthermore, maybe use boxplots for showing the ensemble spread from the three models. Right now, the gray lines just create a lot of "noise".

- Figure 11: Please remove the lines between the dots! You do not show a continuous time-series here!

- Figure 12: Usually, the x-axis in reliability plots shows the Forecast probability and the y-axis the observed relative frequency. Is there any reason why you have not defined the axes like this?

- Table 1: Resolut0ion should be km^2; remove bracket after dampening

**References**

Arnal, L., Cloke, H. L., Stephens, E., Wetterhall, F., Prudhomme, C., Neumann, J., Krzeminski, B., and Pappenberger, F.: Skilful seasonal forecasts of streamflow over Europe?, Hydrol. Earth Syst. Sci., 22, 2057–2072, https://doi.org/10.5194/hess-22-2057-2018, 2018.

Ionita, M., Nagavciuc, V. Forecasting low flow conditions months in advance through teleconnection patterns, with a special focus on summer 2018. Sci Rep 10, 13258 (2020). https://doi.org/10.1038/s41598-020-70060-8

Samaniego, Luis, Stephan Thober, Niko Wanders, Ming Pan, Oldrich Rakovec, Justin Sheffield, Eric F. Wood, Christel Prudhomme, Gwyn Rees, Helen Houghton-Carr, Matthew Fry, Katie Smith, Glenn Watts, Hege Hisdal, Teodoro Estrela, Carlo Buontempo, Andreas Marx, and Rohini Kumar. " Hydrological Forecasts and Projections for Improved Decision-Making in the Water Sector in Europe", Bulletin of the American Meteorological Society 100, 12 (2019): 2451-2472, accessed Feb 22, 2022, https://doi.org/10.1175/BAMS-D-17-0274.1

---

## Author Comment (AC1)

*We thank the reviewer for their careful reading of and thorough comments on our manuscript. In the following, we repeat the reviewers' comments for clarity and added our replies to them in italic font. Additions and changes to the manuscript are indicated by an italic and bold font.*

In general, the study adresses a very recend and highly relevant topic as it discusses the skill of seasonal forecasts for predicting runoff. Besides the "raw" runoff forecasts from HTESSEL, E-HYPE and EFAS, the authors also apply a simple but effective quantile mapping bias correction in order to match the distribitions between observed and forecasted runoff. The bias-corrected forecasts are then compared against observed runoff at a single gauge in the river Rhine at Lobith, that serves as a proxy for inflow to the rive r and lake IJssel. The authors conclude that after bias correction, streamflow forecasts have skill up to four months ahead in spring, when streamflow is dominated by snow melt. During summer, this skillful period decreases to about 1-2 months. While the study focuses on a single basin, the results are still important for the hydrological community.

Major comments

But this limited scope is my main poin of criticism. As far as I can tell, the authors did not run any of the models, but rather used publicly available data. Furthermore, there are multiple data sources for runoff observations across Europe. Thus, while the study is certainly of high relevance for the water management of the IJssel reservoir, it could serve as a state-of-the-art skill assessment for a much larger domain and, hence, would be an update for, e.g., Arnal et al. (2018). None of the applied methods are tailored to the study domain or require any local adjustments or knowledge. Thus, the whole framework could be easily applied to multiple sites and, hence, present the skill of seasonal streamflow forecasts from current hydrological forecasting systems for major European river basins. This would be a very important and much needed contribution to the whole European hydrometeorolgcal community. Therefore, I would suggest to include at lease some more rivers and gauges to a) increase the importance and scope of the study and b) see if the author's findings can be transferred to other river basins.

*We fully agree that there is no reason for our results to be only valid for the Rhine basin, as, indeed, we did not apply any site-specific corrections. However, the manuscript does not aim to be an update to, for example Arnal et al., (2018), as we mainly use the same data, and perform similar analyses as they do (but zooming in on our region of interest). Because we confirmed the results of Arnal et al. (2018) for the Rhine basin, our finding of enhanced predictability of extreme events will most probably also apply to other reservoirs in Europe. It is however, outside of the scope of this paper, to confirm this on the European scale. This would, of course, be very useful future study. We substantially extended the discussion section, where we also addressed this point. For the added text, we refer to page 5 of this letter, where the full text is included.*

Furthermore, other climatic variables like precipitation or temperature are equally important for the reservoir management. If the authors decide to stick to the IJssel reservoir, I would strongly suggest to include some other variables as well to make the whole study a bit more comprehensive.

While this is a quite frequent comment, I also think that the overall language, wording and presentation of the paper could be substantially improved.

*We fully agree that other climatic variables are highly relevant to reservoir management as well. However, from literature and initial analyses it is clear that meteorological forecast skill beyond a few weeks is generally very low in northwestern Europe. In the case of Lake IJssel, other relevant variables are local precipitation, global radiation (which governs local land evaporation and hence local water use) and wind speed (which is an important driver for open water evaporation. All of these variables are known to have very little skill beyond a few weeks (see e.g. Lucatero et al, 2018). We confirmed these results with our data and decided to focus the scope of the manuscript on river discharge as this has the highest forecast skill beyond a few weeks. We already summarized this by including precipitation in one of our skill metrics (CRPSS). In the new modified version, we extended this analysis to include precipitation in other skill metrics also.*

*We have added a remark to the discussion section about meteorological forecast skill, which is nearly absent after a month:*

***We made similar plots to Figure 8 for basin averaged precipitation and temperature compared to E-OBS observations, to investigate meteorological predictability during extreme conditions, and found virtually no skill after the first month. As these findings confirm results from literature (e.g., Lucatero et al., 2018), we do not show all of these results.***

*Lucatero, D., Madsen, H., Refsgaard, J. C., Kidmose, J., and Jensen, K. H.: On the skill of raw and post-processed ensemble seasonal meteorological forecasts in Denmark, Hydrol. Earth Syst. Sci., 22, 6591–6609, 2018.*

*Regarding the readability of the manuscript, we re-read the manuscript and clarified the wording where we deemed appropriate to do so. This affects a number of locations, which we will not all repeat in this review reply.*

Minor comments

Line 61: Add whitespace before "In section 2..."

*Done.*

Lines 88 - 89: Maybe re-phrase to: Total runoff from the HTESSEL land surface scheme is aggregated at native resolution over the Rhine catchment?

*We agree this is a better sentence and adopted this suggestion.*

Line 89: Could you give a reference for this statement? Could the lack of a routing scheme explain the bad results of the raw HTESSEL forecasts?

*The travel time between Switzerland and Lobith is about 5 days (Khanal et al., 2019), hence lack of routing should not significantly affect weekly data. We added this information and reference to the manuscripts:*

**"Because we aggregate the daily time series to weekly averages and the travel time between Switzerland is about 5 days (Khanal et al, 2019), the lack of routing is assumed to have a negligible impact on the results."**

Line 99: similar to, for example, the Variable...

*Done.*

Line 104: The HYPE-model, whose European application is referred to as E-HYPE, is a semidistributed...

*Done.*

Line 106: Whitespace after 215km^2.

*Done.*

Line 125: This is the first time that you discuss the runoff observations that are used in the study. Please add some more details like the source of runoff observations to, e.g., section 2.2.

*We added a short section describing the origin of the observations:*

**All forecast systems are benchmarked against discharge observations from the Dutch national water authorities, disseminated through the open data portal https://waterinfo.rws.nl. Daily discharge observations at Lobith were obtained, spanning the period from 1/1/1901 to 1/1/2019.`**

Lines 149 - 150: How did you compute the CRPSS for precipitation forecasts? Which "reference" did you use?

*Good point. We used basin-averaged E-OBS data as a benchmark for precipitation, and added this information and corresponding reference to the manuscript:*

**"… over the Rhine basin, using spatially averaged EOBS-precipitation v21e (Cornes et al., 2018) as benchmark.***"***

Lines 156 - 158: Remove "three in this case" as this is obvious from the previous sentences.

*Done.*

Line 158: ...is benchmarked using AN observed streamflow climatology...

*Done.*

Lines 161 - 163: This sounds extremely complicated and, to some extend, different to the general definition of the term "forecast resolution". But this might be unintentional and could be improved by simply citing standard references for the evaluation of forecasts.

*The employed metric is indeed different from the general definition of forecast resolution, and resulted from discussions with water managers, who are looking for a practical, and operationally usable, approach to interpret forecast quality. We, therefore, added it to the other metrics. However, we can see the confusion caused by the reference to forecast resolution and now referred to it as the 'absolute difference between the probability of higher-than-normal and lower-than-normal discharge'.*

Line 171: lead tme

*Done.*

Line 175: using

*Done.*

Lines 185 - 192 and Figure 7: I do not know how "substantial" this Diebold-Mariano-Tests really are to your overall asssesment... Especially as you are comparing ensemble medians and, hence, do not consider the full ensemble. I assume that a simple comparsion of forecast-tailored skill-scores (e.g., a generalization of Figure 10) should give more comprehensive information.

*It is certainly true that Figure 7 only considers the forecast median and not the full ensemble. Because we already present a number of skill scores that do include the full forecast ensemble, showing that the differences between forecast systems are typically small and varying between months, we consider it a useful addition to assess their statistical significance. We are not aware of a statistical test that would do this considering the full ensemble, although that would indeed be preferable.*

*It is not completely clear to us what the reviewer means by 'a generalization of Figure 10', which indeed shows month-to-month variation of a forecast-tailored skill score (the RPSS). Calculating different metrics on a month-to-month basis would most probably show a similar variability.*

Line 201: ...QM-factors are derived DURING 2000, 2001, and 2004 - 2015

*Done.*

Line 258: Why are you particularly referring to machine learning approaches here?

*Because there is a human component to the distribution of the Rhine branches (it depends on the management of two specific weirs in the Nederrijn), physical modelling of this distribution is difficult. In our experience, given sufficient historic data availability, machine learning approaches could help here. We added a reference (Suntaramont et al., 2020), where machine learning algorithms were included to model weir management.*

*Suntaranont, B., Aramkul, S., Kaewmoracharoen, M., & Champrasert, P. (2020). Water irrigation decision support system for practical weir adjustment using artificial intelligence and machine learning techniques. Sustainability, 12(5), 1763.*

Section 3.1: The method is ussually known as "Quantile mapping". Please correct this throughout the manuscript. Furthermore, how did you treat the tails of the distribution? Getting these parts of the distributions right is quite important especially as you are also focusing on low-flow events in your evaluation.

*We changed 'matching' into 'mapping' throughout the manuscript. We agree that getting the tails of the distribution right is important. However, corresponding to the cited references, we treated each of the 50 2% bin in entire distribution in the same way. After we inspected Figure 3, we see no reason to adopt a different treatment for the distribution tails.*

Section 3.2: Maybe replace "Forecast skill metrics" with "Evaluation of forecast skill"?

*Done.*

Section 3.2.1: Why did you only use the BS for low-flow forecasts? The skill of high-flow forecasts is quite important as well.

*We agree that high flows are important but they are not the main scope of the paper as they are not relevant for reservoir management, in this particular case at least. We added a figure to the paper showing the Brier score for exceedances of the upper tercile, and thus illustrating the seasonal predictability of high flows.*

Section 5: You do not really "discuss" your findings here, but rather summarize your study and give an outlook. As there are already several publications on this topic (e.g., Arnal et al., 2018; Samaniego et al., 2019; Ionita & Nagavciuc, 2020), it would be interesting to put your findings in the context of these other studies

*We agree with the reviewer that the discussion could be more extensive and thank him/her for the suggestions. We substantially extended the discussion and in effect added a paragraph to the discussion section, discussing the results in a broader context.*

*Our results largely confirmed earlier results by (Arnal et al., 2018), who found increased forecast skill in spring in (partly) snowfed rivers and lowest forecast skill in summer. In winter, teleconnections like NAO positively affect forecast skill in European rivers (Scaife et al., 2019; Bierkens and van den Hurk, 2007). Arnal et al. (2018) indicated few regions in Europe with increased forecast skill in summer, but the Rhine appeared not to be one of them. To ensure statistical robustness, these analyses resulted in average forecast skill over a large number of years. Although we confirmed that average summer forecast skill is low and varies between years, we also found that skill increases with the extremity of the event: summers with extremely low discharges were skillfully forecast longer ahead, up to four months. Ionita and Nagavciuc (2020) found similar results for the summer of 2018 based on statistical forecasting systems. They found especially sea surface temperature in parts of the northern Atlantic ocean to be a good predictor of Rhine river discharge for long lead-times. Meißner et al. (2017) found such statistical forecasting methods to outperform more physically based methods such as SEAS5. The forecast systems used in this study all depend on meteorological forecasts based on the ECMWF model. Recently, multi-model forecast systems have been developed using different general circulation models (GCMs) for atmospheric forecasts (Samaniego et al., 2019; Wanders et al., 2019; Muhammad et al., 2018). This highlights the importance of multi-model forecasting systems incorporating both statistical and physically based methods.*

*Our finding of increasing predictability with event extremity suggests that individual forecasts contain useful information, which could be discarded as noise by statistical analyses. As was also noted by Viel et al. (2016) and Meißner et al. (2017), small forecast skill does not mean forecasts are not useful for decision-makers. Our results indicate that when a large fraction of the forecast ensemble is in the lower or upper tercile, the probability of the forecast being correct is high. This, therefore, is very important information for decision makers in the reservoir management in anticipation of extreme low-flow conditions.*

*Given the identical meteorological forcing, the differences found in our study are presumably caused by the schematization and/or parameterization of the hydrological models. By considering and comparing the relevant processes separately (e.g. groundwater flow, glacier melt) the reasons for these differences could be further explored. The representation of glacial melt, for example, is more sophisticated in E-HYPE compared to Lisflood, which might contribute to slightly higher skill in summer for E-Hype, whereas the opposite may be the case for snow melt, causing slightly higher skills for EFAS in early spring. The absence of hydraulic processes in H-TESSEL is likely to cause overall lower forecast skills. Fully disentangling these differences requires model output per component for all forecast systems, which was not feasible in the current study.*

*We focused on one specific location in the Rhine basin. However, discharge at Lobith nearly integrates the entire Rhine basin, which covers, with 160,000 km2, a substantial part of Western-Europe. In more spatially extensive studies, (e.g. Arnal et al., 2018), results for the Rhine appeared comparable to many other areas in Europe, suggesting that our results are applicable to other areas in Europe. In general, streamflow is more predictable in river systems with long memory due to snow-processes, groundwater contribution and dampening from lakes and reservoirs and groundwater contribution, all of which apply to the Rhine, and less in arid climates with fast hydrological response to precipitation (Pechlivanidis et al., 2020). Our finding of the higher predictability of low flow extremes*

*would, in our view, also translate to other catchments with similar characteristics. This remains to be confirmed in a future study.*

Figure 1: What are the black dots? Increase fontsize of legend, R^2 and y --> hard to read

*The black dots are the locations of the two discharge gauges. We now annotated them (added the names of the locations at the dot locations), and increased the font size of the text in the right panel.*

Figure 2: Please increase font-size for the legend. Furthermore, I would remove "Forecast bias corrections using...." and rather just write "Upper row: CDFs from EFAS, E-HYPE, ... derived from forecasts issued on April 1st for lead months 1 (April), 3 (June), 5 (August) and 7 (October); bottom row: mapping factors between observed and forecasted CDFs".

*We increased the legend font size, and adjusted the caption as suggested.*

Figure 4: Do you evaluate the skill of Apr. to Sept. forecasts from different issue dates (months) or rather the skill of forecasts that have been issued from Apr. to Sept.?

*The first option is right; i.e., we evaluate the forecast for a given month for the different issue dates. For example, in the panel labeled with "April" , we evaluate forecasts issued between the previous October and April 1$^{st}$. We added this information to the caption of Figure 4 (to which figures 5 and 6 refer, as the same applies to them). The caption of Figure 4 now reads:*

**Figure 4. CRPSS for all hindcasts, with and without bias correction, for each lead time (months). Also shown is the forecast of precipitation aggregated over the Rhine basin (see text). For a given calendar month we evaluate forecasts at seven issue dates, so for example in the upper left panel we evaluate forecasts issued between September and March.**

Figure 8: You write that you're using raw HTESSEL-forecasts but in the text (Lines 198 - 201), you write that "we consider the bias-corrected results". Or did I understand something wrong here?

*The caption was incorrect, we indeed used corrected forecasts. We corrected the caption in a new version:*

**Figure 8. Example of a tercile plot using the QM-corrected HTESSEL forecasts at Lobith…**

Figure 9: Increase the "thickness" of the observations as they are very hard to distinguish from all other lines. Furthermore, maybe use boxplots for showing the ensemble spread from the three models. Right now, the gray lines just create a lot of "noise".

*We increase the line thickness and replaced the grey lines that indicated the ensemble spread by a grey area between the minimum and maximum ensemble members.*

Figure 11: Please remove the lines between the dots! You do not show a continuous time-series here!

*We agree and replaced the lines between the dots by a stairtype plot, to make clear the time series is discrete. We considered showing solely points to be not sufficiently visible.*

Figure 12: Usually, the x-axis in reliability plots shows the Forecast probability and the y-axis theobserved relative frequency. Is there any reason why you have not defined the axes like this?

*There is no particular reason to have the axis oriented as they were. As it is indeed more customary to transpose the figures, we did so.*

Table 1: Resolution should be km^2; remove bracket after dampening

*We agree; adjusted as suggested.*

References

Arnal, L., Cloke, H. L., Stephens, E., Wetterhall, F., Prudhomme, C., Neumann, J., Krzeminski, B., and Pappenberger, F.: Skilful seasonal forecasts of streamflow over Europe?, Hydrol. Earth Syst. Sci., 22, 2057-2072, https://doi.org/10.5194/hess-22-2057-2018, 2018.

Ionita, M., Nagavciuc, V. Forecasting low flow conditions months in advance through teleconnection patterns, with a special focus on summer 2018. Sci Rep 10, 13258 (2020). tps://doi.org/10.1038/s41598-020-70060-8

Samaniego, Luis, Stephan Thober, Niko Wanders, Ming Pan, Oldrich Rakovec, Justin Sheffield, Eric F. Wood, Christel Prudhomme, Gwyn Rees, Helen Houghton-Carr, Matthew Fry, Katie Smith, Glenn Watts, Hege Hisdal, Teodoro Estrela, Carlo Buontempo, Andreas Marx, and Rohini Kumar. " Hydrological Forecasts and Projections for Improved Decision-Making in the Water Sector in Europe", Bulletin of the American Meteorological Society 100, 12 (2019): 2451-2472, accessed Feb 22, 2022, https://doi.org/10.1175/BAMSD-17-0274.1

*We thank the reviewer for their careful reading of and thorough comments on our manuscript. In the following, we repeat the reviewers' comments for clarity and added our replies to them in italic font. Additions and changes to the manuscript are indicated by an italic and bold font.*

I would like to thank the editor for the opportunity to review this manuscript.

This study investigates the performance of three continental- / global-scale streamflow forecasting systems (HTESSEL, EFAS and E-Hype) in predicting inflows to Lake Ijssel – a major surface water reservoir in the Netherlands. All three forecasting systems are driven with ECMWF SEAS5 seasonal climate forecasts but differ in their underlying hydrological modelling approach. The authors applied bias correction to streamflow forecasts using Quantile Mapping and subsequently assessed the skill of raw and post-processed forecasts for predicting inflows. A particular focus was placed on dry conditions, i.e. the predictability of low-flow events.

In my view, the overall focus of the study to compare three leading continental- or global-scale streamflow forecasting systems for predicting streamflow at the local-scale is interesting and I believe the study has applications and implications that may be relevant to a wider audience, e.g. how to best translate outputs from continental- or global-scale streamflow forecasts into local-scale applications. However, my main concern relates to the novelty of the study. In its current form, the manuscript is written similar to a Technical Report that describes methods and results of forecast post-processing and verification applied to one single study location, without sufficiently linking it to the research context, and may be of interest to a very local audience. I am missing an attempt to generalise the findings and place them into a broader context and emphasise implications that are applicable in other regions too (using Lake Ijssel inflows as a case study). If the authors addressed this, I believe the study would be of interest to a wider audience.

The study design and methodology (including post-processing and forecast verification) are robust. While more advanced streamflow post-processing methods exist and could be investigated, the authors clearly show an improvement in skill for the study location of interest. The application of multiple verification metrics (continuous ranked probability skill score, mean error, Brier skill score and reliability diagrams) to 23 years of hindcast data, in a cross-validation approach, is appropriate and thorough.

Major comments:

Novelty and relevance to a wider audience: As outlined above, the novelty of the study and relevance to a wider audience is not very clear. I would ask the authors to place the work into a broader research context and interpret the results more broadly, highlighting implications for researchers and practitioners in other regions. While the introduction places the research into a wider context to some extent, there is no clear link between the introduction and the rest of the manuscript which describes the results in a very detailed way, focusing on one individual location. It would be great if the authors could come back to the broader research gap in their discussion and conclusions, and interpret the results more broadly, e.g. what are key messages for the hydrological community who aim to apply regional- or global-scale seasonal streamflow forecasts for individual catchments? How does this study compare to similar systems implemented in other parts of the world? What are advantages and disadvantages of the described approach?

Results and Discussion: The results section is very comprehensive and describes the results of the forecasting approach in a very detailed way, focusing on a range of forecast performance metrics. However, I am missing an interpretation that goes beyond simply describing the results. The discussion section itself is very short and immediately starts with limitations of the study and further research, without an actual discussion and interpretation of the study findings. I would ask the authors to add more discussion of their results – linking them back to their research aims or questions that are outlined in the introduction, placing them into the context of the existing literature and highlighting implications or applications of the findings (followed by limitations and further research, as is already included). Potential points of discussion: What can we learn from these results that is relevant in other regions? Can some of the differences in results be traced back to differences in the underlying specifications of the hydrological models (e.g. consideration of routing) and what does it mean for other locations? Could a multi-model ensemble approach be useful?

Readability: I had difficulties following some sections. The manuscript would benefit from revision with the aim of re-wording sections and sentences to be clearer and more concise. Additionally, there are some typos throughout the text (I have included a few examples under "minor comments" but it applies to the manuscript overall).

*We agree with the reviewer that the discussion could be more extensive and thank also this reviewer for the suggestions. We substantially extended the discussion and in effect added a number of paragraphs to the discussion section, discussing the results in a broader context. We refer to our response to the first reviewer for the added text (see page 7), as he/she made essentially the same remark.*

*Regarding the readability of the manuscript, we re-read the manuscript and clarified the wording where we deemed appropriate to do so. This affects a number of locations, which we will not all repeat in this letter.*

Other comments / suggestions:

Some of the results and figures are very detailed and could be presented in a more concise or synthesised way. A few suggestions and observations in relation to the figures:

Figures 4-6: Could you present the results for each forecasting system (raw and bias corrected outputs) and each lead time, aggregated over all months (April to September), to be able to compare the systems more generally? One possibility could be to add an additional sub plot that presents aggregated results across all months.

*We added a panel to these figures showing the skill scores aggregated over all shown months. Concerning the more general remark about a more concise presentation of the figures: we believe the month-by-month analysis of forecast skill is essential for the storyline. We, therefore, chose to add an extra panel. We also added result for March, to keep an even number of panels.*

Figure 10: Similar as above, it would be great to see the overall RPSS scores aggregated over all years and months, to compare the three systems directly with each other. Would it be possible to add another sub plot that presents the results aggregated across all years and months?

*This figure is different compared to the above as it is intended to show the variability between years. Aggregated over all available years, it would show similar results as were shown in Figures 4-6 (albeit with slightly higher skill as we only consider 'extreme' years). Moreover, we do not have all the relevant years available for E-HYPE, so we cannot directly compare the aggregated results.*

Figure 11: I found it a little unusual to see the years on the x-axis, but in order of dryness / wetness. Would it be possible to use annual streamflow as x-axis (i.e. presenting forecast skill as a function of average streamflow)?

*We agree that discharge is a more logical variable to display on the x-axis. However, we deem it important for the storyline to also show the year numbers. We, therefore, now show the average summer discharge on the x-axis and added a top axis displaying the year numbers.*

Figure 12: It is not clear to me what the right column of Figure 12 shows. It would be good to provide more explanation in the text and/or caption – how could this be interpreted and used?

*We agree that the information in this panel is somewhat unconventional and arose during discussions with local water managers. We rewrote the corresponding text in Sections 3.2.2 and 4.2, as well as the figure caption. We now refer to it as the absolute difference between higher-than-normal and lower-than normal discharge and extended the explanation. What the metric is intended to show is that the reliability of the forecast increases as this absolute difference increases. It is, therefore, related to a reliability plot but appeared to appeal to water managers as it can provide them directly with a probability that a forecast will turn out correct, as a function of the presented absolute difference.*

Minor comments:

P2 L47: Please change "takes" to "take" and remove "also".

*Done.*

P3 L61: Space missing after "as follows."

*Corrected.*

P3 L79: I suggest to change "data forecasting systems" to "seasonal forecasting systems".

*We agree with the suggestion; corrected as suggested.*

P6 L135-136: Please change "is investigated 3.2.2" to "is investigated in Section 3.2.2".

*Done.*

P8 L175: "using" instead of "usiing"

*Done.*